# Regioselective One-Pot Synthesis, Biological Activity and Molecular Docking Studies of Novel Conjugates *N*-(p-Aryltriazolyl)-1,5-benzodiazepin-2-ones as Potent Antibacterial and Antifungal Agents

**DOI:** 10.3390/molecules27134015

**Published:** 2022-06-22

**Authors:** Asma Nsira, Hasan Mtiraoui, Sami Chniti, Hanan Al-Ghulikah, Rafik Gharbi, Moncef Msaddek

**Affiliations:** 1Laboratory of Heterocyclic Chemistry Natural Products and Reactivity/CHPNR, Department of Chemistry, Faculty of Science of Monastir, University of Monastir, Monastir 5000, Tunisia; asma_nsira@yahoo.fr (A.N.); mtiraoui1hasan@gmail.com (H.M.); samichniti@yahoo.fr (S.C.); moncefmsadek@gmail.com (M.M.); 2Department of Chemistry, College of Sciences, Princess Nourah bint Abdulrahman University, P.O. Box 84428, Riyadh 11671, Saudi Arabia; 3Laboratory of Applied Chemistry and Environment, Department of Chemistry, Faculty of Science of Monastir, University of Monastir, Monastir 5000, Tunisia; raf_gharbi@yahoo.fr

**Keywords:** 1,5-benzodiazepin-2-ones, azides, click chemistry, CuAAC, *N*-triazolo-benzodiazepinones, antibacterial activity, antifungal activity, docking

## Abstract

Novel 1,2,3-triazolo-linked-1,5-benzodiazepinones were designed and synthesized via a Cu(I)-catalyzed 1,3-dipolar alkyne-azide coupling reaction (CuAAC). The chemical structures of these compounds were confirmed by ^1^H NMR, ^13^C NMR, HMBC, HRMS, and elemental analysis. The compounds were screened for their in vitro antibacterial and antifungal activities. Several compounds exhibited good to moderate activities compared to those of established standard drugs. Furthermore, the binding interactions of these active analogs were confirmed through molecular docking.

## 1. Introduction

The development of new therapeutic agents is one of the major goals in medicinal chemistry research [1]. Generally, evidence that agents are modulating more than one target may develop a wider field of therapeutic applications compared to single-target drugs [2,3]. Hence, the actual increase in interest in agent discovery is already addressing multiple biological targets for many therapeutic treatments [4,5].

One of the privileged structures that have been recently updated by Patchett et al. [6,7] is the 1,5-benzodiazepine (BZD) derivatives that have been repeatedly reported to display tranquilizing, muscular relaxant, anticonvulsant, hypnotic, and sedative effects [8,9,10]. Actually, the use of this class of scaffolds is not only limited to anxiety and stress conditions but also seemingly minor changes in their structures that can produce a host of different biological activities [11]. Accordingly, polycyclic BZD derivatives **A**, **B**, and **C** have proven their bioactivity against peptides hormone (**A**), interleukin converting enzymes (**B**), and potassium blockers (**C**) [12,13,14] (Figure 1).

Moreover, in previous work, our research group has reported the production and subsequent determination of photoluminescence properties of an understudied family of 1,5-benzodiazepin-2-one derivatives. Furthermore, the recent work published by Chiraz Ismail et al. reports on the synthesis of some fluorescent *N*-triazolo-1,5-benzodiazepine-2-ones [15,16].

Because the *N*-functionalization of benzodiazepines is highly desired for the development of novel powerful molecular targets [17], it appears that the *N*-1,2,3-triazolo-1,5-benzodiazepine scaffold has great importance due to the remarkable biological relevance of such combination [18,19]. Triazoles belong to an important class of heterocycles. They display an ample spectrum of biological activities and are widely employed as pharmaceuticals and agrochemicals [20,21,22]. More particularly, the 1,2,3-triazole derivatives that exhibit favorable physicochemical properties interact with different biological targets through hydrogen bonding and dipole interactions, improving both the potency and specificity of the resulting analogs [23,24].

Thus, and as a continuation of our ongoing research to synthesize novel 1,5-benzodiazepines derivatives bearing a triazole moiety [25], we turn our attention to designing novel hybrid conjugates of 1,2,3-triazoles tethered to 1,5-benzodiazepines namely the *N*-triazolo-1,5-benzodiazepin-2-ones. On the other hand, the click chemistry methodology is one of the most used strategies for simple access to these compounds, particularly the Cu(I)-catalyzed 1,3-dipolar alkyne-azide coupling reaction (CuAAC) [26]. In addition, derivatives **4a**–**i** and **6a**–**c** were evaluated for their antibacterial and antifungal potentials, and further molecular docking of synthesized compound **6** was also performed.

## 2. Results and Discussion

Our synthetic strategy for building the *N*-triazolo-1,5-benzodiazepine scaffolds is based on the CuAAC reaction and involves the preparation of *N*-alklynic benzodiazepine **2a**–**c** reacted with aromatic azides **3a**–**d**. Thus, the key intermediate BZD **2** was primarily prepared following the method of E. Latteman et al. [27,28]. We treated compound **1a**–**c** with propargyl bromide in the presence of sodium hydride as a base in *N*,*N*-dimethylformamide at 0 °C. Eventually, DMF was found to be especially effective in this reaction for weakening the bromine-carbon bond [29]. Under these experimental conditions, the reaction monitored by TLC showed the formation of a single product that was identified, on the basis of its spectral data, as the *N*-prop-2-yn-1,5-benzodiazpin-2-one **2a**–**c**. Note here that compound **2a** has already been prepared by our research team [13]. In addition, H. Ahabchane and co-workers have prepared BZD derivatives but are limited in substitution patterns [30].

The ^1^H NMR spectrum of compound **2a**–**c** recorded at 300 MHz in CDCl_3_ exhibited characteristic signals from which chemical shifts and multiplicities we were able to assign the propargyl group. Thus, for compound **2b**, taken as an example, the spectrum showed a doublet at 4.25 ppm (*J* = 2.4 Hz) corresponding to the methylene group at C-1″ coupled with the acetylenic proton H-3″ which appears as a triplet at 2.30 ppm (*J* = 4.80 Hz).

Taking notes that the propargylation of the benzodiazepine could obviously occur either on the hydroxyl or on the amide function [31], the presence of the deshielded phenolic hydrogen singlet observed at~13.95 ppm excluded from the beginning the obtention of the *O*-prop-2-yn-1,5-benzodiazepin-2-one (Figure 1). Particularly in **2b**, the non-equivalence of the methylenic protons H-3 (a pair of two doublets at 3.00 ppm (*J* = 12.6 Hz) and 4.70 ppm(*J* = 16.8 Hz)) is undoubtedly consistent with partial non-planarity of the heptatomic ring. Similarly, this result is also cited in our previously described *N*-isopropylated-1,5-benzodiazepine-2-one [32,33].

The reluctance of the hydroxyl group to react was rationalized in terms of a steric hindrance due to a strong intermolecular hydrogen bonding between the hydroxyl group (13.95 ppm) and the nitrogen of the imine C=N functionality at the 5-position of the diazepine ring [34].

The analysis of the ^13^C NMR spectra recorded at 75.47 MHz came comforting the obtention of the *N*-propargyl-1,5-benzodiazepinone **2b** that showed a peak at 37.1 ppm (C-1″), 72.0 ppm (C-3″) as well as 78.1 ppm (C-2″) of the propargyl group.

The corresponding azides **3a**–**c** were prepared according to the reported method via a devastation reaction of *p*-substituted aniline using NaNO_2_ and a diluted solution of HCl in ethanol at 0 °C followed by treatment with NaN_3_ [35].

The coupling of azides **3a**–**c** and the *N*-propargyl-1,5-benzodiazepin-2-ones **2a**–**c** was carried out in DCM at room temperature using CuI as catalyst and triethylamine as an additive base. Very interesting pentacyclic compounds **4a**–**I** were then isolated in suitable yields (Figure 2). The reaction parameters were optimized using the *N*-propargyl-1,5-benzodiazepines **2a**, the azides **3c**, and the CuI as catalysts. The reaction did occur whatever the solvent used. Replacing acetonitrile with toluene increased the yields owing to a better solubility of the starting materials (entries 2 and 4). The reaction resulted in comparable yields when performed at room temperature or under gentle heating. On the other hand, an increase in the amount of the catalyst (from 5 to 10 mol%) did not modify the yield (entries 6 and 7).

However, the excess of CuI probably caused a decrease in the performance due to the deposition of copper species on the dipole and the low solubility of cuprous iodide in triethylamine (entries 8). DCM proved to be by far the most suitable solvent at room temperature (entries 6) (Table 1)

Thus, one can state that the use of 1 m mole *N*-propargyl-1,5-benzodiazepine **2a**–**c**, aromatic azide **3a**–**c** (2 eq) at rt for 4 h in DCM as a solvent with CuI (5 mol%) as catalyst and triethylamine (2.5 eq) as an additive [36] are the best experimental conditions to generate the series of new *N*-triazolyl-1,5-benzodiazepin-2-one **4a**–**i** in suitable yield (Figure 2 (Table 2)).

In particular we have observed that the solubility of the 1,2,3-triazole-BZD conjugates **4a**–**i** is enhanced in most of the organic solvents. This may be attributed to the new functionalities present in these novel conjugates.

Unambiguous proofs for the obtained products **4a**–**i** were obtained from their ^1^H/^13^C NMR and 2D NMR spectra, which were consolidated by HRMS and elemental analysis (see Appendix A).

As mentioned in the introduction, the effects of such benzodiazepines on the nervous system are abundantly described in the literature [37]. Moreover, interesting biological activities are observed with some analog derivatives, but their very low hydrosolubility can restrict their applications. Generally, when glycopyranosyl is attached to the nitrogen of the heptatomic ring systems, it can increase the water solubility and confer amphiphilic properties.

Obviously, there is no single function for oligosaccharides. Perhaps their most important function is to serve as recognition markers. Additionally, oligosaccharides have the ability to alter the intrinsic properties of the molecules to which they are attached [38].

Accordingly and encouraged by the above interesting result, we have extended this method to the synthesis of novel *N*-galactopyranosyl-*N*-triazolo-1,5-benzodiazepines. Therefore, we screened an azido galactpyranosyl [39], a choice that was not fortuitous insofar as our research team has used it for the synthesis of some optically active pyrazolines [40,41,42,43].

As exemplified in (Figure 3), the reaction proceeded smoothly to completion, and the corresponding *N*-galactopyranosyl-*N*-triazolo-1,5-benzodiazepinones products **6a**–**c** were obtained after 8 h with excellent yields and with high purity (Table 3).

To find the optimal experimental conditions for the reaction, the cycloaddition reaction was firstly carried out in different solvents: DCM, acetonitrile, toluene, and a mixture of DMF/H_2_O at room temperature and under reflux. Finally, it was found that DMF/H_2_O(8:2) was the most suitable solvent (Table 4).

The use of both dimensional and bidimensional NMR spectroscopy techniques allowed one to deduce unambiguously the exclusive formation of the regioisomeric species, namely the 1,4-triazoles (as exemplified for **6b**).

Four singlets integrating three hydrogens each and corresponding to the methyl of the galactopyranose part appeared at 1.34, 1.36, 1.47, and 1.50. A peak integrating one proton was also observed at 7.69 ppm and assigned as the characteristic H-5″triazolic hydrogen.

A minor influence of the triazole group was observed on the proton H-1″ in front of the triazole ring at C-1, which has a chemical shift of 4.25 ppm in the precursor **2b** and a two doublet at 4.75 ppm and 5.09 ppm in **6b**. Furthermore, a modest downfield shift was observed for the galactose H-6′′′ signal due to the influence of the triazole ring at C-4, changing from 3.55 ppm in azide**5** to approximately 4.52 ppm in the triazole products for **6** series.

The resulting 1,4-regioisomers were evidenced by the presence in the NOESY spectrum of an NOE between the triazolic proton H-5″ and the H-5′′′ proton of the galactopyranosyl moiety, in addition to another NOE between the proton H-5′′′ and H-6′′′. Such regiospecificity agrees with that cited in the literature [44].

To our knowledge, the obtention of these *N*-galactopyranosyl-*N*-triazolo-1,5-benzodiazepinones conjugates **6a**–**c** is very demanded given the interesting pharmacological properties of some analogs so far reported by I. Carvalho et al. As a matter of fact, they proved to be moderate to weak TcTS (Trypanosoma cruzi and its cell surface trans-sialidase) inhibitors in vitro [45].

Most prepared 1,5-benzodiazepin-2-ones were evaluated for antibacterial and antifungal activity in order to survey the possible biological activities of this class of compounds [46,47].

### 2.1. Biological Activity

#### 2.1.1. Antibacterial Activity

Were tested in vitro for antibacterial activity against an array of eight bacteria using streptomycin as a control, with the findings expressed as MIC in g/mL. (Table 5). The obtained data revealed that all the tested compounds **4a**–**i** showed suitable inhibition against all strains. Particularly compounds **4d** (R_1_ = Me, R_2_ = H) and **4e** (R_1_ = Me, R_2_ = OMe) in series 1 might be the major active compounds, and they all showed a similar activity potential, especially against *S. epidermidis* (MIC = 32 µg/mL) showing values better than the reference antibiotic. Most of the tested compounds displayed poor activity against *E. coli* and *S. typhimurum*. Further, toward *B. cereus*, derivatives **4e** seems to contribute better (MIC = 32 μg/mL) than the other analogs followed by **4d** (MIC = 64 μg/mL) whereas, against *S. aureus*, compound **4f** (R_1_= Me, R_2_= NO_2_) displayed the highest activity (MIC= 32 μg/mL). Toward *M. luteus*, derivative **4e** was found to be the most active compound, followed by **4d** and **4g**. Moreover, compounds **4d** then **4e** due to hydrogen atom and methoxy group in the phenyl para-position, respectively, showed the best values for the antibacterial activity compared to other analogs against *E. fecalis*. Furthermore, toward *L. monocytogenes*, also **4d** and **4e** displayed noticeable antibacterial activity. On the other hand, as depicted in Table 5, the obtained data demonstrate that all the tested compounds **6a**–**c** showed better values of the antibacterial potential compared to compounds **4a**–**i**. Finally, these findings clearly showed the importance of the added fragments to the 1,5-benzodiazepine **1** via the methylene linker to confer activity, essentially the nature of the aromatic system and the galactopyranosyl attached to the triazole ring in the activity.

#### 2.1.2. Antifungal Activity

The target compounds **4a**–**i** and **6a**–**c** were assayed for inhibitory activity against clinically important pathogenic fungi such as the *Candida albicans* and the *Aspergillus flavus.* Ketoconazole was used as the reference drug (Table 6). All the titled compounds showed good to moderate inhibition against the tested fungal pathogens. Particularly, compound **4d** (R_1_ = Me, R_2_ = H) revealed excellent activity against both the *Candida albicans* and the *Aspergillus flavus.*

Furthermore, the tested compounds **6a** (R = H) and **6b** (R = OMe) showed suitable activity against *Aspergillus flavus* with MIC = 64 µg/mL, which was significantly more potent than *Ketoconazole.* Toward *Candida albicans*, **6b** (MIC = 32 µg/mL) seems to be the most active, followed by its analogs **6a** and **6c** (R = Cl). These obtained results suggest that the galactopyranosyl part on the C-4 triazole ring of compound **6b** is favorable for enhancing antifungal activity.

### 2.2. Molecular Docking Studies

A molecular docking study of the newly synthesized compound of series 1 (**4a**–**i**) and series 2 (**6a**–**c**)was conducted to gain insights into its probable mechanism of action. Indeed, the crystallized structure of Staphylococcus epidermidis TcaR in complex with streptomycin (PDB code: 4EJW) was taken as the target receptor, and the binding pocket was validated by performing redocking of the ligand (Streptomycin). The binding pocket and the interaction of the ligand in complex with the target receptor are shown in Figure 2. Molecular docking calculations of all the test compounds were carried out with Auto Dock vina software. The docked ligand with the lowest binding free energy was used for analysis in Table 7.

As can be seen from the results, the molecular docking for the representative compounds: the most active derivative in series 1 is BZD **4e**, the most active derivative in series 2 is compound **6c**, and the redocked «streptomycin» showed that the ligands were well oriented toward the active site gorge. Thus, **4e** formed a conventional hydrogen bond with GLN-B-61 through its hydroxyl group besides a Pi-Donor hydrogen bond with GLN-A-31. In addition, the ligand **4e** was oriented to a hydrophobic pocket composed of ALA-A-24 and ALA-A-38 with Pi-Alkyl interactions. The methylbenzodiazepine ring contributed to shaping interaction with HIS-A-42 and Alkyl interaction with VAL-A-63. The methoxytriazole moiety formed Pi-Alkyl interactions with ALA-B-24 and ALA-B-38 besides a stacking interaction with HIS-B-42 (Figure 2B,B′).

On the other hand, ligand **6c** set up H-bonds with ASN-B-20 and HIS-A-42 through its *N*-galactopyranosyl and BZD pharmacophores, respectively. This finding demonstrates the crucial role of the *N*-galactopyranosyl in series 2 (compounds **6a**–**c**) linked to the triazole ring, which took the place of the aryl group in series 1(compounds **4a**–**i**). Furthermore, **6c** formed some interesting Alkyl interactions with residues: VAL-A-63, ALA-B-24, LEU-B-27, ALA-B-38, and HIS-B-42 via its *N*-galactopyranosyl fragment, which displayed a Pi-Sigma interaction with HIS-B-42. In addition, derivative **6c** showed Amide-Pi stacked with SER-A-41 and hydrophobic Pi-Alkyl and Alkyl interactions with VAL-B-63. (Figure 2C,C′).

From these results, it can be inferred that docked compound, especially derivative **6c**, probably showed its antibacterial activity in a similar way as that of the Streptomycin antibiotic (Figure 2A,A′) by interfering with the functioning of epidermidis TcaR in complex with streptomycin receptor.

## 3. Materials and Methods

### 3.1. Instruments and Methods

Toluene and methylene chloride (DCM) were obtained from MBRAUN′s MB SPS-800 apparatus and dried according to conventional protocols. Unless otherwise specified, cyclohexane, ethyl acetate (EtOAc), acetonitrile (CH_3_CN), and diethyl ether (OEt_2_) were acquired in ACS-grade quality and utilized without additional purification. Unless otherwise noted, commercially available reagents were utilized without further purification.

^1^H and ^13^C NMR spectra were recorded with an AC-300 Bruker spectrometer with tetramethylsilane as an internal reference. Chemical shifts are reported in parts per million. Two-dimensional NMR experiments were performed with an Avance-300 Bruker spectrometer. Multiplicities are described as s (singlet), d (doublet), dd, dd, etc. (doublet of doublets), t (triplet), and m (multiplet). High-resolution mass spectra of compounds **4b**, **4e**, and **4g** were performed within a Hewlett-Packard 5890/5970 GC mass spectrometer. Elemental analysis was recorded on a PERKIN–ELMER 240B microanalyzer.

All the reactions were followed by TLC using aluminum sheets of Merck silica gel 60 F254, 0.2 mm. The spots were visualized through illumination with a UV lamp (λ = 254 nm) and/or staining with KMnO_4_. Column chromatography purifications were performed on silica gel (40–63 μm) carried out on Merck DC Kiesel gel 60 F-254 aluminum sheets. The starting material **1a**–**c** was prepared according to the literature [8]. Melting points of benzodiazepines **2a**–**c**, **4a**–**l**, and **6a**–**c** were determined on a Buchi 510 capillary melting point apparatus.

### 3.2. Synthesis of N-Propargyl-1,5-benzodiazepinones *(**2a**–**c**)*

NaH (60% in mineral oil, 0.88 g, 2.4 mmol, 1.2 equiv.) was added to a solution of 4-(2′-hydroxypheny1)-1,5-benzodiazepin-2-one **1a**–**c** (2 mmol) in DMF (15 mL) at 0 °C under nitrogen. Before the mixture was stirred for 10 to 15 min and propargyl bromide, 1.2 equiv was added. The reaction mixture was maintained at room temperature for 6 h. The reaction mixture was kept at room temperature. The raw ingredient was poured into distilled water, and dichloromethane was used to extract it. The organic layers were mixed together and dried over anhydrous MgSO_4_, then filtered and concentrated under reduced pressure. The crude substance was purified using silica gel column chromatography (80:20 hexane/EtOAc).

#### 3.2.1. 1-prop-2-ynyl-4-(2-Hydroxyphenyl)-3H-1,5-benzodiazepin-2-one (**2a**)

This compound was preparedaccordingtotheliteraturemethod [13].

#### 3.2.2. 1-prop-2-ynyl-4-(2-Hydroxyphenyl)-8-methyl-3H-1,5-benzodiazepin-2-one (**2b**)

Yield 377 mg (62%). Yellow solid, m.p. 136–138 °C. ^1^H NMR (300 MHz, CDCl_3_) δH 2.30 (t, 1H, H-3″), 3.00 (d, 1H, H-3b, *J* = 12.3 Hz), 4,25 (d, 2H, CH2-1″, *J* = 2.4 Hz), 4.70 (d, 1H, H-3a, *J* = 17.1 Hz), 6.93 (t, 1H, H-5′), 7.01 (d, 1H, H-3′, *J* = 7.5 Hz), 7.15 (dd, 1H, H-7), 7.23 (d, 1H, H-6, *J* = 3.9 Hz), 7.40 (t, 1H, H-4′), 7.51 (s, 1H, H-9), 7.83 (d, 1H, H-6, *J* = 6.6 Hz), 13.95 (s, 1H, OH). ^13^C NMR (75.47 MHz, CDCl_3_) δC 20.7 (CH_3_-8a), 37.0 (C-3), 38.1 (C-1″), 72.0 (C-3″), 78.4 (C-2″), 117.7 (C-3′), 118.6 (C-1′), 121.0 (C-5′), 121.7 (C-9), 126.5 (C-7), 126.7 (C-4′), 127.7 (C-6), 131.7 (C-6′), 133.4 (C-9a), 135.5 (C-5a), 137.7 (C-8), 161.6 (C-2′), 163.5 (C-2), 164.5 (C-4). Anal. Calcd for C_19_H_16_N_2_O_2_ (304.12): C, 74.98; H, 5.30; N, 9.20 found: C, 74.90; H, 5.27; N, 9.18.

#### 3.2.3. 1-prop-2-ynyl-4-(2-Hydroxyphenyl)-8-chloro-3H-1,5-benzodiazepin-2-one (**2c**)

Yield 452 mg (60%). Yellow solid, m.p 176–178 °C. ^1^H NMR (300 MHz, CDCl_3_) δ_H_2.37 (t, 1H, H-3″), 3.03 (d, 1H, H-3a, *J* = 12.3 Hz), 4.27 (d, 2H, CH_2_-1″, *J* = 2.4 Hz), 4.74 (d, 1H, H-3b, *J* = 17.1 Hz), 6.98 (t, 1H, H-5′), 7.04 (d, 1H, H-3′, *J* = 8.4 Hz), 7.34 (d, 1H, H-7, *J* = 8.7 Hz), 7.44 (t, 1H, H-4′), 7.46 (s, 1H, H-9), 7.69 (d, 1H, H-6,*J* = 8.7 Hz), 7.89 (d, 1H, H-6′, *J* = 7.8 Hz), 13.74 (s, 1H, OH).^13^C NMR (75.47 MHz, CDCl_3_) δ_C_20.7 (C-8a), 37.0 (C-3), 38.1 (C-2″), 72.0 (C-4″), 78.4 (C-3″), 117.7 (C-3′), 118.6 (C-1′), 121.0 (C-5′), 121.7 (C-9), 126.5 (C-7), 126.7 (C-8), 127.7 (C-6), 131.7 (C-6′), 134.0 (C-4′), 135.5 (C-9a), 137.7 (C-5a), 161.6 (C-2′), 163.5 (C-2), 164.5 (C-4).Anal. Calcd for C_18_H_13_ClN_2_O_2_ (324.07): C, 66.57; H, 4.03; N, 8.63 found: C, 66.01; H, 4.12; N, 8.69.

### 3.3. General Procedure for the Synthesis of Compounds *(**4a**–**i**)*

CuI (5.0 mg, 0.025 mmol, 5 mol percent) and the corresponding phenyl azide **3a**–**e** derivative were added to a mixture of compounds **2a**–**c** (0.5 mmol, 1 eq) and Et_3_N (2.0 eq, 134 l, 1 mmol) in DCM (20 mL) (1.0 mmol, 2.0 eq).At room temperature, the reaction mixture was stirred for 4 h. The filtrate was concentrated under reduced pressure after the crude reaction was filtered using Celite^®^. Flash column chromatography on silica gel (Cyclohexane/EtOAcfrom 100:0 to 90:10) was usedtopurifythecrudesubstance, yielding pure 4a–oin 77–89% yields**.**

#### 3.3.1. 4-(2-Hydroxyphenyl)-1-((1-phenyl)-1H-1,2,3-triazol-4-yl)methyl)-1,5-benzodiazepin-3H-2-one (**4a**)

Yield 178 mg (87%). Yellow solid, m.p. 201–203 °C.^1^H NMR (300 MHz, CDCl_3_) δH 3.00 (d, 1H, H-3a, *J* = 12.00 Hz), 4.25 (d, 1H, H-3b, *J* = 2.4 Hz), 4.90 (d, 1H, H-1a″, *J* = 14.7 Hz), 5.25 (d, 1H, H-1b″, *J* = 15 Hz), 6.96 (t, 1H, H-7), 7.05 (d, 1H, H-6, *J* = 8.4 Hz), 7.32 (d, 1H, H-9, *J* = 7.5 Hz), 7.41–7.45 (m, 4H, H-4′, H-5′, H-4′′′, H-8), 7.48 (t, 2H, H-3′′′, H-5′′′), 7.70 (d, 2H, H-2′′′, H-6′′′, *J* = 7.5 Hz), 7.86 (d, 1H, H-3′, *J* = 8.1 Hz), 8.11 (s, 1H, H-5″), 8.13 (d, 1H, H-6′, *J* = 8.1 Hz).^13^C NMR (75.47 MHz, CDCl_3_) δC 38.2 (C-3), 44.8 (C-1″), 117.9 (C-1′), 118.3 (C-6), 119.1 (C-2′′′, C-6′′′), 120.4 (C-6′), 122.5 (C-5″), 126.1; 126.7; 127.6; 134.2 (C-4′, C-5′, C-8, C-4′′′), 128.8 (C-3′), 129.3 (C-3′′′, C-5′′′), 129.7 (C-9a), 135.2 (C-5a), 136.9 (C-1′′′), 138.2 (C-4″), 162.1 (C-2′), 164.9 (C-4), 165.1 (C-2). Anal. Calcd for C_24_H_19_N_5_O_2_ (409.45): C, 70.40; H, 4.68; N, 17.10; found: C, 70.22; H, 4.37; N, 17.16.

#### 3.3.2. 4-(2-Hydroxyphenyl)-1-((1-(4-metoxyphenyl))-1H-1,2,3-triazol-4-yl)methyl)-1,5-benzodiazepin-3H-2-one (**4b**)

Yield 189 mg (86%). Yellow solid, m.p. 174–176 °C.^1^H NMR (300 MHz, CDCl_3_) δH 3.02 (d, 1H, H-3a, *J* = 12.00 Hz), 3.88 (s, 3H, OCH3), 4.27 (d, 1H, H-3b, *J* = 12.00 Hz), 4.96 (d, 1H, H-1a″, *J* = 15.00 Hz), 5.27 (d, 1H, H-1b″, *J* = 15.30 Hz), 6.98–7.09 (m, 4H, H-arom), 7.34–7.49 (m, 4H, H-arom), 7.62 (d, 2H, H-3′′′, H-5′′′, *J* = 7.80 Hz), 7.88 (d, 1H, H-arom, *J* = 8.1 Hz), 8.05 (s, 1H, H-5″′), 8.16 (d, 1H, H-6′, *J* = 8.10 Hz), 13.95 (1s, 1H, OH). ^13^C NMR (75.47 MHz, CDCl3) δC21.08 (CH_3_-4a′′′), 38.2 (C-3), 44.8 (C-1″), 55.6 (OCH_3_), 114.7 (C-arom), 118.0 (C-1′), 118.3 (C-arom), 119.1 (C-arom), 122,0 (C-2′′′), 122.6 (C-5″), 123.3 (C-arom), 126.1 (C-arom), 126.9 (C-arom), 127.6 (C-arom), 129.3 (C-arom), 130.4 (C-9a), 134.1 (C-arom), 135.3 (C-5a), 138.2 (C-1′′′), 144.2 (C-4″) 159.8 (C-4′′′), 162.2 (C-2′), 164.0 (C-4), 165.0 (C-2). Anal. Calcdfor C_25_H_21_N_5_O_2_ (439.16): C, 68.33; H, 4.82; N, 15.94; found: C, 68.12; H, 4.89; N, 16.06. HRMS (ESI+): calcd. for C_25_H_21_N_5_NaO_2_[M+Na]^+^: 460.1749; found: 460.1763.

#### 3.3.3. 4-(2-Hydroxyphenyl)-1-((1(4-nitrophenyl)-1H-1,2,3-triazol-4-yl)methyl)-1,5-benzodiazepin-3H-2-one (**4c**)

Yield 165 mg (78%). Yellow solid, m.p. 225–227 °C. ^1^H NMR (300 MHz, CDCl_3_) δH 3.04 (d, 1H, H-3a, *J* = 12 Hz), 4.28 (d, 1H, H-3b, *J* = 12 Hz), 5.05 (d, 1H, H-1a″, *J* = 15 Hz), 5.26 (d, 1H, H-1b″, *J* = 15,3 Hz), 6.99 (t, 1H, H-arom), 7.08 (d,1H, H-arom, *J* = 8.10 Hz), 7.34–7.50 (m, 4H, H-arom), 7.88 (d, 1H, H-arom, *J* = 7.80 Hz), 7.95 (d, 2H, H-3′′′, H-5′′′, *J* = 9.00 Hz), 8.07 (d, 1H, H-6′, *J* = 8.10 Hz), 8.19 (s, 1H, H-5″), 8.41 (d, 2H, H-2′′′, H-6′′′, *J* = 9.00 Hz).^13^C NMR (75.47 MHz, CDCl_3_) δ= 38.2 (C-3), 44.6 (C-1′′), 117.9 (C-1′), 118.3 (C-arom), 119.2 (C-arom), 120.4 (C-arom), 122.3 (C-arom), 123.1 (C-5″), 125.5 (C-arom), 126.3 (C-arom), 127.0 (C-arom), 127.6 (C-arom), 129.3 (C-arom), 134.3 (C-9a), 134.9 (C-1′′′), 138.4 (C-5a), 141.0 (C-4″), 147.3 (C-4′′′) 162.2 (C2′), 164.1 (C-4), 165.2 (C-2). Anal. Calcd for C_24_H_18_N_6_O_4_ (454.14): C, 63.43; H, 3.99; N, 18.49; found: C, 63.15; H, 4.09; N, 18.23.

#### 3.3.4. 4-(2-Hydroxyphenyl)-8-methyl-1-((1-phenyl)-1H-1,2,3-triazol-4-yl)methyl)-1,5-benzodiazepin-3H-2-one (**4d**)

Yield 173 mg (82%). Yellow solid, m.p. 202–204 °C. ^1^H NMR (300 MHz, CDCl_3_) δH 2.40 (s, 3H, CH_3_-8a), 2.99 (d, 1H, H-3a, *J* = 12.00 Hz), 4.23 (d, 1H, H-3b, *J* = 12.30 Hz), 4.91 (d, 1H, H-1a″, *J* = 15.30 Hz), 5.22 (d, 1H, H-1b″, *J* = 15.30 Hz), 6.94 (t, 1H, H-arom), 7.03 (d, 1H, H-arom, *J* = 8,4 Hz), 7.22 (s, 1H, H-9), 7.39 (t, 2H, H-arom), 7.48 (t, 2H, H-arom), 7.69 (d, 2H, H-arom, *J* = 8,4 Hz), 7.84 (dd, 1H, H-arom, *J* = 7,8 Hz), 7.96 (d, 1H, H-arom, *J* = 8.4 Hz), 8.07 (s, 1H, H-5″). ^13^C NMR (75.47 MHz, CDCl_3_) δC 20.7 (CH_3_-8a), 38.2 (C-3), 44.7 (C-1″), 118.2 (C-1′), 119.1 (C-arom), 118.1 (C-arom), 120.4. (C-arom), 122.4 (C-5″), 123.0 (C-arom) 126.9 (C-arom), 128.6 (C-arom), 128.7 (C-arom), 129.3 (C-arom), 129.7 (C-arom), 132.8 (C-9a), 134.0 (C-arom), 135.1 (C-5a), 136.1 (C-8), 138.0 (C-1′′′), 144.5 (C-4″), 162.2 (C-2′), 163.8 (C-4), 164.9 (C-2). Anal. Calcd for C_25_H_21_N_5_O_2_ (423.17): C, 70.91; H, 5.00; N, 16.54; found: C, 70.51; H, 5.09; N, 16.24.

#### 3.3.5. 4-(2-Hydroxyphényl)-8-méthyl-1-((1-métoxyphényl)-1H-1,2,3-triazol-4-yl)méthyl)-1,5-benzodiazépin-3H-2-one (**4e**)

Yield 187 mg (83%). Yellow solid, m.p. 184–186 °C.^1^H NMR (300 MHz, CDCl_3_) δH 2.40 (s, 6H, CH_3_-8a), 2.98 (d, 1H, H-3a, *J* = 12.00 Hz), 3.85 (s, 3H, OCH_3_-4′′′), 4.22 (d, 1H, H-3b, *J* = 12.30 Hz), 4.92 (d, 1H, H-1a″, *J* = 15.00 Hz), 5.21 (d, 1H, H-1b″, *J* = 15.30 Hz), 6.94–7.05 (m, 4H, H-arom), 7.18 (m, 1H, H-arom), 7.21 (s, 1H, H-9), 7.39 (t, 1H, H-arom), 7.58 (d, 2H, H-2′′′, H-6′′′, *J* = 9.00 Hz), 7.84 (d, 1H, H-6′, *J* = 8.10 Hz), 7.97 (s, 1H, H-5″), 13.95 (1s, 1H, OH). ^13^C NMR (75.47 MHz, CDCl_3_) δC 19.7 (CH_3_-8a), 20.0 (CH_3_-4a′′′), 38.2 (C-3), 44.6 (C-1″), 117.0 (C-1′), 117.2 (C-arom), 118.0 (C-arom), 121.5 (C-arom), 122.3 (C-5″), 125.9 (C-arom), 126.9 (C-arom), 128.2 (C-arom), 129.4 (C-arom), 131.8 (C-arom), 133.0 (C-arom) 135.1 (C-9a), 137.0 (C-5a), 143.3 (C-4″), 158.8 (C-1′′′), 161.2 (C-4′′′), 162.3 (C-2′), 162.8 (C-4), 163.9 (C-2). Anal. Calcd for C_26_H_23_N_5_O_3_ (453.18): C, 68.86; H, 5.11; N, 15.44; found: C, 69.12; H, 5.29; N, 15.46; HRMS (ESI+): calcd. for C_26_H_24_N_5_O_3_ [M+H]^+^: 454.1879; found: 454.1879.

#### 3.3.6. 4-(2-Hydroxyphenyl)-8-methyl-1-((1-(4-nitrophenyl)-1H-1,2,3-triazol-4-yl)methyl)-1,5-benzodiazépin-3H-2-one (**4f**)

Yield 185 mg (79%). Yellow solid, m.p. 235–237 °C. ^1^H NMR (300 MHz, CDCl_3_) δH 2.41 (s, 6H, CH_3_-8a), 3.01 (d, 1H, H-3a, *J* = 12.00 Hz), 4.23 (d, 1H, H-3b, *J* = 12.00 Hz), 5.03 (d, 1H, H-1a″, *J* = 15.00 Hz), 5.24 (d, 1H, H-1b″, *J* = 15.00 Hz), 6.95 (t, 1H, H-arom), 7.00 (d, 1H, H-arom, *J* = 8.10 Hz), 7.16 (m, 1H, H-arom), 7.22 (s, 1H, H-9), 7.41 (t, 1H, H-arom), 7.80 (m, 2H, H-arom), 7.90 (d, 2H, H-2′′′, H-6′′′, *J* = 9.30 Hz), 8.12 (s, 1H, H-5″), 8.37 (d, 2H, H-3′′′, H-5′′′, *J*J = 9.00 Hz).^13^C NMR (75.47 MHz, CDCl_3_) δC 19.7 (CH_3_-8a), 37.2 (C-3), 43.5 (C-1″), 117.3 (C-1′), 118.1 (C-arom), 119.4 (C-arom), 121.1 (C-arom), 121.8 (C-5″), 124.4 (C-arom), 125.9 (C-arom), 127.6 (C-arom), 128.2 (C-arom), 131.5 (C-arom), 133.1 (C-9a), 135.4 (C-5a), 137.2 (C-8), 140.0 (C-1′′′), 144.4 (C-4″), 146.2 (C-4′′′), 161.2 (C-2′), 162.9 (C-4), 164.0 (C-2). Anal. Calcd for C_25_H_20_N_6_O_4_ (468.15): C, 64.10; H, 4.30; N, 17.94; found: C, 63.85; H, 4.19; N, 17.64.

#### 3.3.7. 4-(2-Hydroxyphényl)-8-chloro-1-((1-phényl)-1H-1,2,3-triazol-4-yl)méthyl)-1,5-benzodiazépin-3H-2-one (**4g**)

Yield 173 mg (78%). Yellow solid, m.p. 246–248 °C. ^1^H NMR (300 MHz, CDCl_3_) δH 2.99 (d, 1H, H-3a, *J* = 12.30 Hz), 4.29 (d, 1H, H-3b, *J* = 12.30 Hz), 4.85 (d, 1H, H-1a″, *J* = 13.50 Hz), 5.27 (d, 1H, H-1b″, *J* = 14.10 Hz), 6.99 (t, 1H, H-arom), 7.07 (d, 1H, H-arom, *J* = 8.40 Hz), 7.41–7.50 (m, 4H, H-arom), 7.53 (t, 1H, H-arom), 7.74 (d, 2H, H-2′′′, H-6′′′, *J* = 9.00 Hz), 7.88 (d, 1H, H-arom, *J* = 7.20 Hz), 8.19 (s, 1H, H-5″), 13.64 (1s, 1H, OH). ^13^C NMR (75.47 MHz, CDCl_3_) δC 38.4 (C-3), 44.7 (C-1″), 117.8 (C-1′), 118.4 (C-arom), 119.3 (C-arom), 120.5 (C-arom), 124.6 (C-arom), 126.4 (C-arom), 127.6 (C-arom), 128.9 (C-arom), 129.4 (C-arom), 129.8 (C-arom), 131.3 (C-8), 133.9. (C-4″), 134.5 (C-arom), 139.2 (C-4′′′), 162.2 (C-2′), 164.6 (C-4), 164.9 (C-2). Anal. Calcd for C_24_H_18_ClN_5_O_2_ (443.11): C, 64.94; H, 4.09; N, 15.78; found: C, 65.24; H, 3.89; N, 16.08.HRMS (ESI+): calcd. for C_24_H_18_BrClN_5_O_2_[M+Br]^+^: 522.1324; found: 522.1324.

#### 3.3.8. 4-(2-Hydroxyphenyl)-8-chloro-1-((1-metoxyphényl)-1H-1,2,3-triazol-4-yl)methyl)-1,5-benzodiazepin-3H-2-one (**4h**)

Yield 191 mg (81%). Yellow solid, m.p. 236–238 °C.^1^H NMR (300 MHz, CDCl_3_) δH 2.99 (d, 1H, H-3a, *J* = 12.30 Hz), 3.89 (s, 3H, OCH3-4a′′′), 4.28 (d, 1H, H-3b, *J* = 12.30 Hz), 4.86 (d, 1H, H-1a″, *J* = 15.00 Hz), 5.25 (d, 1H, H-1b″, *J* = 15.00 Hz), 6.98–7.09 (t, 4H, H-arom), 7.28 (dd, 1H, H-arom, *J* = 8.7 Hz), 7.37 (s, 1H, H-9), 7.44 (t, 1H, H-arom), 7.63 (d, 2H, H-2′′′, H-6′′′, *J* = 9.00 Hz), 7.86 (d, 1H, H-arom, *J* = 8.10 Hz), 8.05 (s, 1H, H-5″), 8.28 (d, 1H, H-arom, *J* = 7,8 Hz), 13.80 (1s, 1H, OH). ^13^C NMR (75.47 MHz, CDCl_3_) δC 37.2 (C-3), 43.8 (C-1″), 54.6 (OCH_3_-4a′′′), 113.7 (C-arom), 116.8 (C-1′), 117.3 (C-arom), 118.2 (C-arom), 121.0 (C-arom), 122.2 (C-5″), 125.4 (C-arom), 127.0 (C-arom), 128.3 (C-arom), 129.3 (C-arom), 131.8 (C-8), 133.3 (C-9a), 134.9 (C-1′′′), 135.8 (C-4″), 158.8 (C-4′′′), 161.1 (C-2′), 163.1 (C-4), 163.6 (C-2). Anal. Calcd for C_25_H_20_ClN_5_O_2_ (473.13): C, 63.36; H, 4.25; N, 15.29; found: C, 65.07; H, 4.46; N, 14.78.

#### 3.3.9. 4-(2-Hydroxyphényl)-8-chloro-1-((1-phényl)-1H-1,2,3-triazol-4-yl)méthyl)-1,5-benzodiazépin-3H-2-one (**4i**)

Yield 178 mg (77%). Yellow solid, m.p.>250 °C. ^1^H NMR (300 MHz, CDCl_3_) δH 3.01 (d, 1H, H-3a, *J* = 12.00 Hz), 4.29 (d, 1H, H-3b, *J* = 12.30 Hz), 4.91 (d, 1H, H-1a″, *J* = 15.30 Hz), 5.25 (d, 1H, H-1b″, *J* = 15.00 Hz), 7.00 (t, 1H, H-arom), 7.07 (d, 1H, H-arom, *J* = 8.10 Hz), 7.38 (dd, 1H, H-arom, *J* = 9.00 Hz), 7.46 (m, 2H, H-arom), 7.86 (dd, 1H, H-arom, *J* = 8.10 Hz), 7.97 (d, 2H, H-3′′′, H-5′′′, *J* = 9.00 Hz), 8.09 (d, 1H, H-arom, *J* = 8.70 Hz), 8.24 (s, 1H, H-5′′), 8.42 (d, 2H, H-2′′′, H-6′′′, *J* = 9.00 Hz), 13.64 (1s, 1H, OH). ^13^C NMR (75.47 MHz, CDCl_3_) δC 38.3 (C-3), 44.6 (C-1″), 117.6 (C-1′), 118.4 (C-arom), 119.3 (C-arom), 120.5 (C-arom), 122.6 (C-arom), 124.4 (C-arom), 125.5 (C-arom), 126.5 (C-arom), 127.6 (C-arom), 129.4(C-arom), 131.5 (C-8), 133.6 (C-9a), 134.7 (C-H), 139.3 (C-1′′′), 140.9 (C-4″), 147.3 (C-4′′′), 162.2 (C-2′), 164.8 (C-4), 164.9 (C-2). Anal. Calcdfor C_24_H_17_ClN_6_O_4_ (488.10): C, 58.96; H, 3.51; N, 17.19; found: C, 59.26; H, 3.41; N, 17.00.

### 3.4. General Procedure for the Synthesis of Compounds *(**6a**–**c**)*

CuI (5.0 mg, 0.025 mmol, 5 mol percent) and the suitable galactopyranose azide **5** (1 mmol, 2 eq,) were added to a combination of compounds **2a**–**c** (0.5 mmol, 1 eq) and Et_3_N (2 eq, 134 µL, 1 mmol) in DMF/H2O (8/2). For 8 h, the reaction mixture was stirred at room temperature. The raw material was put into distilled water and extracted with dichloromethane after being filtered through Celite^®^. Flash column chromatography on silica gel (Cyclohexane/EtOAcfrom 100:0 to 90:10) was used to purify the crude substance, yielding pure **6a**–**c** in 80–85% yields.

#### 3.4.1. 4-(2-Hydroxyphényl)-1-(1-(3aR, 5R, 5aS, 8aS, 8bR)-2,2,7,7-tetraméthyltetrahydro-3aH-bis([1,3]dioxolo)[4,5-b:4′,5′-d]pyran-5-yl)méthyl)-1H-1,2,3-triazolo-4-yl)méthyl)-1H-1,5-benzodiazépin-2-one (**6a**)

Yield 202 mg (85%). Yellow solid, m.p. 156–158 °C. ^1^H NMR (300 MHz, CDCl_3_) δH 1.29; 1.37; 1.42; 1.50 (s, 3H, CH_3_), 2.97 (d, 1H, H-3a, *J* = 12.00 Hz), 4.12 (m, 3H, H-3b, H-3′′′, H-5′′′), 4.29 (m, 1H, H-2′′′), 4.45 (m, 1H, H-4′′′), 4.56 (m, 2H, CH2-6′′′), 4.86 (d, 1H, H-1a″, *J* = 15.00 Hz), 5.21 (d, 1H, H-1b″, *J* = 15.30 Hz), 5.51 (s, 1H, H-1′′′), 6.96–8.19 (m, 8H, H-arom), 8.06 (s, 1H, H-5′′), 13.95 (1s, 1H, OH). ^13^C NMR (75.47 MHz, CDCl_3_) δC23.9; 24.3; 25.4; 31.0 (CH_3_-Sucre), 37.7 (C-3), 44.0 (C-1″), 50.3 (C-6′′′), 61.5 (C-2′′′), 67.1 (C-3′′′), 70.3 (C-4′′′), 70.7 (C-5′′′), 96.2 (C-1′′′), 108.5 (C-isop), 109.4 (C-isop), 117.5 (C-1′), 117.7 (C-3′), 118.2 (C-arom), 119.0 (C-arom), 123.2 (C-arom), 125.8 (C-arom), 126.9 (C-arom), 127.5 (C-arom), 128.8 (C-quat), 129.3 (C-arom), 133.5 (C-quat), 134.0 (C-arom), 134.8 (C-8) 137.9 (C-4″), 161.7 (C-2′), 163.8 (C-4), 164.2 (C-2). Anal. Calcd for C_30_H_33_N_5_O_7_ (575.24): C, 62.60; H, 5.78; N, 12.17; found: C, 63.75; H, 6.36; N, 11.68.

#### 3.4.2. 4-(2-Hydroxyphényl)-8-méthyl-1-(1-(3aR, 5R, 5aS, 8aS, 8bR)-2,2,7,7-tetraméthyltetrahydro-3aH-bis([1,3]dioxolo)[4,5-b:4′,5′-d]pyran-5-yl)méthyl)-1H-1,2,3-triazolo-4-yl)méthyl)-1H-1,5-benzodiazépin-2-one (**6b**)

Yellow solid, yield 247 mg (84%), m.p.140–142 °C. ^1^H NMR (300 MHz, CDCl_3_) δH 1.34; 1.36; 1.47; 1.50 (s, 3H, CH_3_), 2.41 (s, 3H, CH_3_-8a), 2.95 (d, 1H, H-3a, *J* = 12.00 Hz), 4.12 (m, 3H, H-3b, H-3′′′, H-5′′′), 4.25 (m, 1H, H-2′′′), 4.36 (m, 1H, H-4′′′), 4.52 (m, 2H, CH_2_-6′′′), 4.75 (d, 1H, H-1a″, *J* = 15.00 Hz), 5.09 (d, 1H, H-1b″, *J* = 15.30 Hz), 5.40 (s, 1H, H-1′′′), 6.87–7.95 (m, 7H, H-arom), 7.69 (s, 1H, H-5″), 13.95 (1s, 1H, OH). ^13^C NMR (75.47 MHz, CDCl_3_)δC 14.1(CH_3_-8a), 24.4; 24.9; 25.9; 31.5 (CH_3_-Sucre), 38.2 (C-3), 44.8 (C-1″), 50.2 (C-6′′′), 66.8 (C-2′′′), 67.1 (C-3′′′), 70.2 (C-4′′′), 70.7 (C-5′′′), 96.2 (C-1′′′), 109.9 (C-isop), 109.9 (C-isop), 118.0 (C-1′), 118.2 (C-3′), 119.0 (C-5′), 123.0 (C-arom), 126.7 (C-5″), 126.9 (C-9), 127.0 (C-9a), 129.3 (C-6′), 133.9 (C-arom), 134.0 (C-5a), 135.9 (C-8) 137.9 (C-4′′), 162.2 (C-2′), 163.8 (C-4) 164.9 (C-2). Anal. Calcd for C_31_H_35_N_5_O_7_ (589.25): C, 63.15; H, 5.98; N, 11.88; found: C, 62.65; H, 5.49; N, 12.15.

#### 3.4.3. 4-(2-Hydroxyphényl)-8-chloro-1-(1-(3aR, 5R, 5aS, 8aS, 8bR)-2,2,7,7-tetraméthyltetrahydro-3aH-bis([1,3]dioxolo)[4,5-b:4′,5′-d]pyran-5-yl)méthyl)-1H-1,2,3-triazolo-4-yl)méthyl)-1H-1,5-benzodiazépin-2-one (**6c**)

Yield 240 mg (80%). Yellow solid, m.p. 172–174 °C. ^1^H NMR (300 MHz, CDCl_3_) δ_H_1.32; 1.39; 1.45; 1.52 (s, 3H, CH_3,_), 2.97 (d, 1H, H-3a, *J* = 12.00 Hz), 4.17 (m, 3H, H-3b, H-3′′′, H-5′′′), 4.32 (m, 1H, H-2′′′), 4.49 (m, 1H, H-4′′′), 4.59 (m, 1H, CH_2_-6′′′), 4.86 (d, 1H, H-1a″, *J* = 15.00 Hz), 5.21 (d, 1H, H-1b″, *J* = 15.30 Hz), 5.54 (s, 1H, H-1′′′), 6.98–8.21 (m, 8H, H-arom), 8.08 (s, 1H, H-5″), 13.95 (1s, 1H, OH). ^13^C NMR (75.47 MHz, CDCl_3_) δ_C_23.8; 24.4; 25.2; 29.9 (CH_3_-Sucre), 37.7 (C-3), 42.0 (C-1″), 50.2 (C-6′′′), 60.3 (C-2′′′), 66.0 (C-3′′′), 70.1 (C-4′′′), 70.4 (C-5′′′), 96.0 (C-1′′′), 107.5 (C-isop), 108.3 (C-isop), 117.5 (C-1′), 117.7 (C-3′), 120.3 (C-arom), 121.0 (C-arom), 123.9 (C-arom), 126.5 (C-arom), 127.8 (C-arom), 128.3 (C-arom), 128.8 (C-quat), 129.7 (C-arom), 133.5 (C-quat), 134.5 (C-arom), 134.8 (C-8) 137.9 (C-4″), 161.7 (C-2′), 163.8 (C-4) 164.2 (C-2).Anal. Calcd for C_30_H_32_ClN_5_O_7_ (609.20): C, 59.06; H, 5.29; N, 11.48; found: C, 58.65; H, 6.01; N, 12.00.

### 3.5. Bioactivity

The in vitro antimicrobial activity of the structurally promising **4a**–**I** and **6a**–**c** against Gram-positive (*B. subtilis* and *S. aureus*…) and Gram-negative (*E. coli* and *P. aeruginosa*…) bacteria were investigated using microdilution assays along with reference drug streptomycin for comparison. H_2_O was used as a negative control.

#### 3.5.1. Antibacterial Tests

##### Microbial Inhibitory Concentration

Microdilution assay The MICs of the compounds were determined by microdilution [48] using standard inocula of 2 × 10^6^ CFU/mL. Serial dilutions of the test compounds were prepared in DMSO. A bacterial fluid (1 mL of 0.5 McFarland standard) was added to each tube. The MIC was visually determined after incubation for 18 h at 37 °C.

#### 3.5.2. Antifungal Activity

The antifungal activity of compounds **4a**–**I** and **6a**–**c** was tested against two fungal species, namely: *Aspergillus flavus* and *Candida albicans*. These fungi were obtained from the (Department of Clinical biology, Laboratory of Analysis, Treatment and valorization of Pollutants of the Environment and Products, Faculty of Pharmacy of Monastir).They were cultured at 25 °C on potato dextrose agar (PDA) medium one week before use.

#### 3.5.3. Molecular Docking Procedure

Molecular docking simulations were performed by Auto Dock 4.2 program package [49]. The optimization of all the geometries of compounds was carried out using ACD (3D viewer) software (http://www.filefacts.com/acd3d-viewer-freeware-info, accessed on 25 March 2022). The three-dimensional structure of PDB (PDB: 4EJW) was obtained from the RSCB protein data bank [50]. First, the water molecules were eliminated, and the missing hydrogens and Gasteiger charges were added to the system during the preparation of the receptor input file. Then, AutoDock Tools were used for the preparation of the corresponding ligand and protein files (PDBQT). Subsequently, pre-calculation of the grid maps was performed using Auto Grid to save much time during docking. Next, the docking calculation was carried out using a grid per map with 40 × 40 × 40 A˚ points of (PDB: 4EJW) in addition to a grid-point spacing of 0.375 A, ˚ which was centered on the receptor in order to determine the active site. The visualization and analysis of interactions were performed using Discovery Studio 2017R2 (https://www.3dsbiovia.com/products/collaborative-science/biovia–discovery-studio/, accessed on 25 March 2022).

## 4. Conclusions

In our study, novel conjugates *N*-triazolo-1,5-benzodiazepinones **4a**–**i** and **6a**–**c** were designed and synthesized. In fact, we have incorporated 1,2,3-triazole at the first position of the heptatomic ring with either linkage employing the Cu(I)-catalyzed 1,3-dipolar alkyne-azide coupling reaction (CuAAC). Compounds synthesized by this method are of high quality, allowing for simple purification and screening in a high throughput manner. Some of them were screened for their antimicrobial activity and have shown good to moderate antibacterial and antifungal activities. Even though the inhibition levels are only at µM levels, we believe these novel classes of *N*-triazolo-1,5-benzodiazepin-2-ones could find applications in biology. Our strategy, therefore, lays the foundations for the future exploration of more potent and selective *N*-fonctonalized-1,5-benzodiazepinones. To understand the mechanism of antibacterial activity and binding mode of these novel derivatives inside the binding pocket of the crystallized structure of *Staphylococcus epidermidis* TcaR in complex with streptomycin and to confirm the experimental results, molecular docking studies were performed.

## Data Availability

Not applicable.

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
