# Peer review of "Regioselective One-Pot Synthesis, Biological Activity and Molecular Docking Studies of Novel Conjugates N-(p-Aryltriazolyl)-1,5-benzodiazepin-2-ones as Potent Antibacterial and Antifungal Agents"

_molecules, 2022, doi:10.3390/molecules27134015_

Round 1

Reviewer 1 Report

In present work synthesis and biological activity properties of novel 1,5-benzodiazepinones were carried out. 

First of all, authors should improve the supplementary materials. The authors should provide spectra at a higher resolution. It is also required to give spectra for all obtained compounds. In addition, why do not the authors provide information about the IR spectra?

Authors should more carefully checked the main text of manuscript. As far as I know, Molecules journal uses citation style like [1], [2], [3] etc and not [i], [ii], [iii] etc. Typos must be corrected. For example 'N-fonctionalization' to 'N-functionalization', 'sodium hydrure' to 'sodium hydroxide', etc. Additionally, in some cases authors used '1H', in some other cases authors used '1H'. I suppose, it is necessary to convert everything to one form.

After performing appropriate corrections, the work can be accepted for publication.

Author Response

Response to review report 1

  • The NMR spectra realized for all the obtained compounds were given in the supplementary materials as the reviewer request. Concerning the information about the IR and the HRMS analysis we could not response affirmatively to this request because we do not have sufficient quantities to do all the analysis and the biological activities. Additionally, it is worth to note that this work was realized in 2015 as a part of my Ph D research and now, I am not allowed to do any synthesis or any analysis given that I have completed my doctoral thesis and that I am no longer part of the research team of the lab.

  • As recommended, we checked more carefully the main text of manuscript.
  • The citation [i], [ii], [iii] etc. were changed in to [1], [2], [3] etc. as requested by the reviewer
  • In all the manuscript we used '1H' and 13C instead of 1H and 13C respectively. We convert everything to one form.

Reviewer 2 Report

The manuscript entitledRegioselective one-pot synthesis, biological activity and molecular docking studies of novel conjugates N-(p-aryltriazolyl)-1,5-benzodiazepin-2-ones as potent antibacterial and antifungal agents”  is written transparently, the synthesis schemes are aesthetic and understandable. Experiments are carried out correctly and well described. Regarding the experimental part, the authors could add an MS analysis for all new compounds (there is only an HR MS analysis for 3 compounds).

The authors performed a very thorough analysis of NMR, assigned protons and carbons to appropriate signals, respectively, and including MS analysis would give a very good and complete documentation of the structure of the new compounds.

The manuscript looks worse in terms of editing. In many places, spaces, upper or lower indices are missing. In many passages, compounds numbers are not written in bold. I marked some of the errors in the manuscript and I add as an attachment.

After corrections, in my opinion, the manuscript may be accepted for printing in Molecules.

Author Response

Response to review report 2

  • We are deeply convinced that MS analysis would give a very good and complete documentation of the structure of the new compounds but unfortunately it was not possible to do the MS spectroscopy for all the compound as we explained above.
  • In many places, spaces, upper or lower indices are corrected., the numbers of compounds were written in bold as the reviewer request.

Reviewer 3 Report

This study describes regioselective one-pot synthesis of novel conjugates of 1,5-benzodiazepin-2-ones as potent antibacterial and antifungal agents. This topic is of interest to the readers and the results are well-presented. However, to be acceptable for publication, I recommend the following amendments:

1) The English language needs careful revision and correction. For example, in line 58, the sentence starting with "On the other hand .." is not complete and needs modification, line 69, the word "sodium hydrure" is not correct, line 95, the word "groupment" is not correct, and many other examples.

2) There is no need for the detailed description of the spectral data inside the main text of the manuscript. For example, the discussion starting at lines 146 and 195 are not necessary and can be moved to the supporting information.

3) The authors did not provide a rationale for docking the prepared compounds into the Staphylococcus epidermidis TcaR. Docking into other important microbial targets should also be examined.

4) The compound in figure 2A needs to be colored by atom types.

5) The number 13 in "13C" should be superscript and the J (for coupling) should be italic.

6) The references listed at the end follow a different numbering system from citations within the text.

Author Response

Response to review report 3

  • The English language were carefully revised and doing the correction. demanded by the reviewer for example, in
  • Line 58, the sentence starting with "On the other hand .." were modified,
  • Line 69, the word "sodium hydrure" were changed to sodium hydride,
  • Line 95, the word "groupment" were corrected, and many other examples were changed.
  • The detailed description of the spectral data inside the main text of the manuscript. For example, the discussion starting at lines 146 and 195 were moved to the supporting information as recommended by the reviewer
  • Interesting note from the reviewer and the justification for the choice of the receptor is as follows: Among all the bacterial strains used in the in vitro part in this work, the only bacterium is Staphylococcus epidermidis TcaR (pdb: 4EJW) (receptor downloaded from the pdb codes site) which is complexed with "streptomycin".( the streptomycin is the antibiotic already used in the in vitro test in part: ‘2.1 Biological activity, 2.1.1 Antibacterial activity’), no other bacteria are found taking the "streptomycin" into its cavity to use it to enrich the modeling part.
  • Figure 2A is changed as the reviewer requested. The software used only outputs the cyan color of any molecule to be colored by atom types.
  • The number 13 in "13C" were superscripted and the type of J (for coupling) was changed to italic.
  • The references listed at the end were changed and follow the numbering system from citations within the text.

Round 2

Reviewer 1 Report

All necessary corrections are made. Thus, manuscript is suitable for publication.